# Comparison of Radar Signatures from a Hybrid VTOL Fixed-Wing Drone and Quad-Rotor Drone

**Jiangkun Gong** [1], **Deren Li** [1], **Jun Yan** [1,*], **Huiping Hu** [2] and **Deyong Kong** [1,3]

1   State Key Laboratory of Information Engineering in Surveying, Mapping and Remote Sense, Wuhan University, Wuhan 430072, China; gjk@whu.edu.cn (J.G.); drli@whu.edu.cn (D.L.); kdykong@hbue.edu.cn (D.K.)
2   Wuhan Geomatics Institute, Wuhan 430079, China; huhuiping@whu.edu.cn
3   School of Information and Communication Engineering, Hubei University of Economics, Wuhan 430205, China
*   Correspondence: yanjun_pla@whu.edu.cn; Tel.: +86-027-6877-8527

**Abstract:** Current studies rarely mention radar detection of hybrid vertical take-off and landing (VTOL) fixed-wing drones. We investigated radar signals of an industry-tier VTOL fixed-wing drone, TX25A, compared with the radar detection results of a quad-rotor drone, DJI Phantom 4. We used an X-band pulse-Doppler phased array radar to collect tracking radar data of the two drones in a coastal area near the Yellow Sea in China. The measurements indicate that TX25A had double the values of radar cross-section (RCS) and flying speed and a 2 dB larger signal-to-clutter ratio (SCR) than DJI Phantom 4. The radar signals of both drones had micro-Doppler signals or jet engine modulation (JEM) produced by the lifting rotor blades, but the Doppler modulated by the puller rotor blades of TX25A was undetectable. JEM provides radar signatures such as the rotating rate, modulated by the JEM frequency spacing interval and the number of blades for radar automatic target recognition (ATR), but also interferes with the radar tracking algorithm by suppressing the body Doppler. This work provides an a priori investigation of new VTOL fixed-wing drones and may inspire future research.

**Keywords:** radar signature; hybrid vertical take-off and landing (VTOL) fixed-wing drone; quad-rotor drone; micro-Doppler; JEM signals

## 1. Introduction

Recently, the hybrid vertical take-off and landing (VTOL) fixed-wing drone entered the drone market. It looks similar to a fixed-wing drone but has a couple of lifting motors, as in multirotor drones. Thus, it can achieve long flight endurance and range, and it can take-off and land vertically, even hovering in the air [1]. Several cases in 2021 revealed the enormous application potential of VTOL fixed-wings. On 15 July 2021, Spanish police seized a large VTOL fixed-wing drone with a wingspan of 4.5 m capable of carrying 150 kg of cargo over 7 h of autonomous flight that was used to carry drugs from Morocco [2]. On 22 July, a Russian aircraft manufacturer unveiled a cutting-edge unmanned aerial vehicle (UAV), the ZALA VTOL, at the MAKS 2021 International Aviation and Space Salon [3]. On 23 August 2021, the US Naval Air Warfare Center Aircraft Division (NAWCAD) contracted a drone company to supply three of its vertical take-off and landing (VTOL) drones for its Blue Water Maritime Logistics UAS program [4]. The Chinese logistics group SF Express has used a new high-capacity cargo-carrying UAV based on the Nuuva V300 model developed by Pipistrel since 2020 [5]. Such VTOL fixed-wing drones have a bright future.

There has been little research investigating the radar detection of VTOL drones. Recently, many academic cases have focused on radar detection, classification, and tracking of radar echoes from drones [6–9]. Their radar cross-section (RCS) values were measured in microwave anechoic chambers and outdoors [10–13]. The radar signals of these drones

were also investigated for automatic target recognition (ATR) [14–19]. Some drone radar detection systems and counter-drone solutions have also been studied [20–22], such as X-band FMCW radar [23,24], Ku-band FMCW radar [25], W-band millimeter wave radar [26], X-band CW radar [27], L-band radar [7], L-band staring radar [28], k-band FMCW radar [29], X-band airborne weather radar [30], multi-static radar [31], and wide/ultra-wide-band radar systems [32]. Yet, most of the targeted drones in these studies are multi-rotor drones, mainly quad-rotor, few are fixed-wing, and even fewer are VTOL drones.

NATO has defined drone categories and classes based on weight, operational altitude, mission radius, and payload [21], including micro, mini, and small, as shown in Table 1. Usually, these multi-rotor drones are consumer-grade micro or mini drones. As far as we know, there was one study that dealt with radar signals of a very small VTOL drone [33], but it was still mini, according to the specifications in Table 1. This kind of drone is probably easy to access and to manipulate. In this paper, we investigated a small industry-tier VTOL drone, based on NATO's categories.

**Table 1.** NATO definitions of drone categories and classes.

| Class | 1 | 1 | 1 |
|---|---|---|---|
| Category | micro | mini | small |
| Weight (kg) | <2 | 2–20 | 20–150 |
| Operational altitude (m) | 90 | 1000 | 1500 |
| Mission radius (km) | 5 | 25 | 50–100 |
| Payload (kg) | <0.5 | 50–100 | 5–50 |

Note: Table only shows drones in class 1.

This paper demonstrates the first radar detection results of a VTOL fixed-wing drone with four lifting rotors. Section 2 introduces the specific characteristics of a typical VTOL fixed-wing drone, TX25A, and a classic quad-rotor drone, DJI Phantom 4. The TX25A has four lifting rotors and one puller rotor, and the DJI Phantom 4 has four lifting rotors. Radar detection, classification, and target tracking are related to radar signatures including RCS, Doppler speed, micro-Doppler signals, and traces. In this paper, we also collected radar data of the TX25A and DJI Phantom 4 and analyzed the above radar signatures. The detected drone results are presented in Section 3, along with a comparison of radar signatures, including radar amplitude, signal-to-clutter (SCR) values, and micro-Doppler signatures. Section 4 discusses the differences and similarities between the radar echoes of TX25A and DJI Phantom 4. Finally, in Section 5, we state the conclusion of this research.

## 2. Materials and Methods

### 2.1. Drones

The specific model of the VTOL fixed-wing drone is TX25A, manufactured by Harryskydream Inc. in Beijing, China. A photo of the TX25A [1] is shown in Figure 1. Physically, it looks like an ordinary fixed-wing drone with four rotors attached to the wings. It has the advantage of the endurance flight of a fixed-wing drone with the ability to hover, putting it in a new hybrid category, and it can also take-off and land vertically like a multirotor drone. The general flight procedure is also a hybrid of multirotor and fixed-wing drone methods. In the take-off stage, the four rotors rotate to provide lifting force. The puller rotor accelerates when the drone is hovering in the air, flying forward while the wing rotors rotate. Thereby, it can take off, land vertically, and hover like a multirotor drone and fly forward with high flight velocity like a fixed-wing drone.

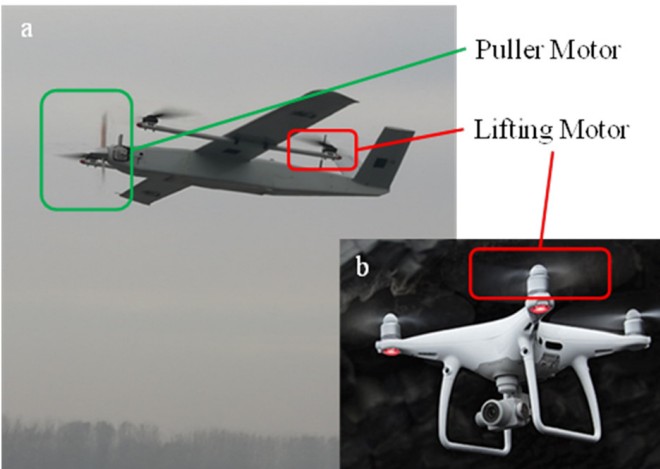

**Figure 1.** Comparison of two drones: (**a**) TX25A, (**b**) DJI Phantom 4. (Photo of TX25A is from http://www.harryskydream.com, accessed on 26 April 2022; photo of DJI Phantom 4 is from https://www.dji.com/, accessed on 11 March 2022).

In order to compare the radar echoes of TX25A with those of a multirotor drone, we also deployed a consumer product, DJI Phantom 4. This is a very common quad-rotor drone on the market and in the literature, provided by the well-known DJI Inc. in China. It has four lifting rotors to provide both lifting force and forward force. The technical parameters of both drones are shown in Table 2. According to NATO's categories, TX25A is a small drone and DJI Phantom 4 is a mini drone. Compared to TX25A, DJI Phantom 4 is much cheaper, much smaller, and more convenient to operate, but has lower flight performance. The reference RCS value of a DJI Phantom 4 is at a level of about 0.01 m$^2$.

**Table 2.** Technical parameters of tested drones.

| Drone Type | Multi-Rotor | Hybrid VTOL Fixed-Wing |
|---|---|---|
| Model | Phantom 4 | TX25A |
| Manufacturer | DJI Inc. | Harryskydream Inc. |
| Take-off weight | 1.38 kg | 26 kg |
| Body size | 0.40 m | 1.97 m |
| Wing span | 0.40 m | 3.60 m |
| Rotor number | 4 | 5 |
| Blade length | 20 cm | 30 cm |
| Max cruise speed | 72 km/h | 115 km/h |

*2.2. Radar Signatures*

Theoretically, the RCS values and Doppler speed of TX25A will be much larger than those of DJI Phantom 4. The Doppler velocity of a drone mainly depends on its flying speed, which is related to its radial velocity. Their cruise speeds also indicate that the Doppler velocity of TX25A would be approximately two times that of DJI Phantom 4 when sharing the same fight route in a test. The RCS of an object primarily depends on the scattering region and the target size [34]. In the optical region, where the wavelength is much smaller than the object size, the RCS of a target seems to be proportional to the target size, which is also sensitive to the target's attitude. However, when the size of an object is similar to the wavelength, it is in the Mie region. Furthermore, the resonance effect at the incident frequency for the target causes the RCS to fluctuate over time. Multiple Doppler peaks in the spectrum are related to the different material compositions of the target. According to the radar equation, the received power of a target is calculated by:

$$P_r = \frac{P_t G^2 \lambda^2 \sigma}{(4\pi)^3 R^4} \tag{1}$$

where $P_t$ is the transmit power, $R$ is the detection range, $G$ is the antenna gain, $\lambda$ is the wavelength, and $\sigma$ is the RCS of the target. As such, the amplitude of radar signals from an object is proportional to the RCS of the object when other factors are the same. Since the physical size of TX25A is approximately four times that of DJI Phantom 4, its RCS will be several times larger.

The radar echoes of both TX25A and DJI Phantom 4 should have typical micro-Doppler signals, also called jet engine modulation (JEM), modulated by the rotating blades. JEM is the well-known micro-Doppler signature appearing in radar signals of aircraft [35] that is used for recognizing aircraft. JEM will produce some spectral peaks with certain adjacent intervals but at a similar-sized amplitude, and each peak corresponds to a blade, while an adjacent interval is the function of the number of blades and rotating speed [36]. When the number of blades is $N$, the group of JEM Doppler frequencies is given by [35,37–39]:

$$f_{md,k} = \frac{L}{\lambda} \Omega cos\beta cos(\varphi_{0,k}), k = 1, 2, 3 \ldots N \tag{2}$$

where $R_0$ is the distance from the radar to the origin of the reference coordinates, $L$ is the blade length, $\lambda$ is the wavelength, $\Omega$ is the rotation rate, $\alpha$ is the azimuth angle, $\beta$ is the elevation angle, and $\varphi_0$ is the initial rotation angle of the blade. JEM appears in a spectrum as a group of spectral Doppler peaks with certain adjacent intervals but at a similar-sized amplitude [40]. When the targeted aircraft is on course and its speed is constant, the JEM frequency spacing interval follows the relation [36]:

$$\Delta f = \frac{N}{\tau} \tag{3}$$

where $N$ is number of blades and $\tau$ is the rotation period. Since both TX25A and DJI Phantom 4 have lifting rotors, their spectra will have JEM signatures. In addition, their JEM frequency spacing will be similar because both the rotating speed and number of lifting blades are similar.

The noted difference may come from the puller rotors of TX25A, which will produce an extra peak around the body Doppler peak because of different dependencies on the aspect angle. Theoretically, only the JEM Doppler is different from the body Doppler, in that it can be separated from the body Doppler. The modulo of the difference vector between the body Doppler and the micro-Doppler is given by:

$$f_{\Delta d} = \left| \overline{f_{bd}} - \overline{f_{md,k}} \right| \tag{4}$$

where $f_{bd}$ is the body Doppler and $f_{md,k}$ is the JEM Doppler. When the flight direction of the lifting blades is parallel to the ground plane, the body Doppler is also parallel to the ground plane, and then $f_{\Delta d}$ can be sufficiently large to separate the micro-Doppler from the body Doppler. However, when the flight direction of the puller blades is perpendicular to the ground plane, $f_{\Delta d}$ is smaller, and sometimes too small to separate. Furthermore, there are fewer puller blades than lifting blades, making the micro-Doppler appear in the spectrum. This means that the JEM signals can sometimes disappear in radar echoes from fixed-wing drones with only pusher blades and in similar situations with pusher blades, such as with hybrid VTOL fixed-wing drones.

### 2.3. Experimental Conditions

The drone data were collected in a coastal area. Figure 2 presents the detection environment and test flowchart. In October 2020, we deployed a radar on the roof of a building near the Yellow Sea in China for several days. The sea during the test was calm, and the sea scale was lower than class 3. The radar wave scanned the sea horizontally. Figure 2a shows the scanning area on the radar PPI image, where the blue line indicates the center of the radar beam. The detection algorithm used was the constant false alarm rate (CFAR). Figure 2b shows the range-Doppler images of the radar data in 20 range gates,

and the time-frequency data of the TX25A and sea clutter. The ATR module of the radar extracted the signatures and identified the drone from the clutter. Moreover, we also used an electro-optical and infrared sensor (EO/IR) camera to track the object and verify the synchronistic detection results of the radar. As soon as the radar captured an object, the optical and infrared camera images were synchronously recorded, and then it could be confirmed that the object was our drone rather than clutter. Figure 2c presents an optical photo and infrared image of TX25A. Finally, the radar tracked the target and recorded the tracking data. The whole flowchart is summarized in Figure 2d.

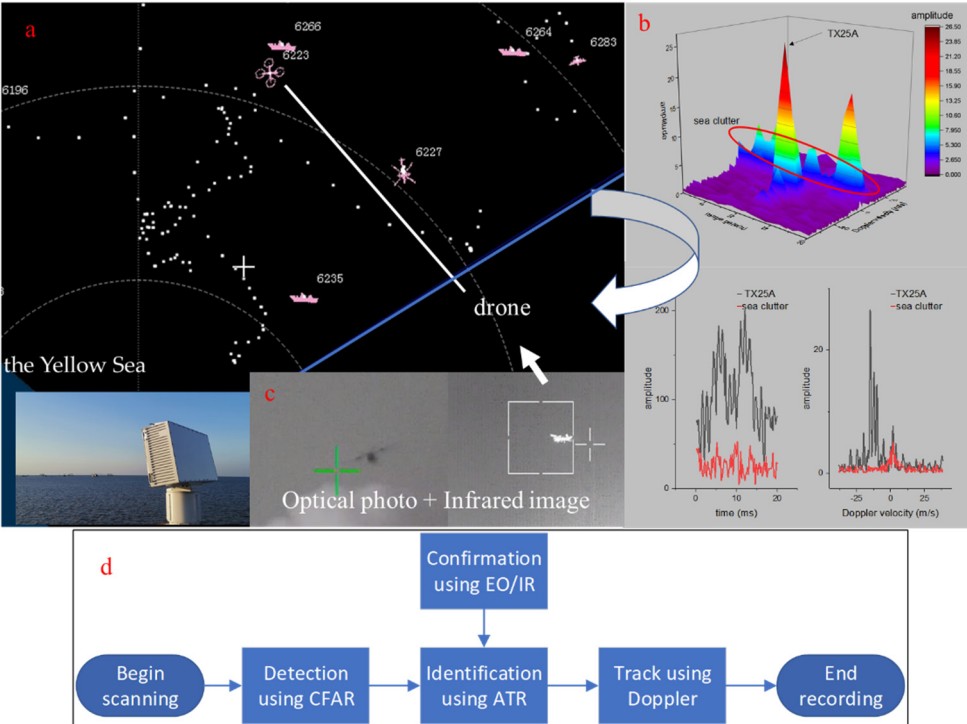

**Figure 2.** Detection of drones using a X-band radar in a coastal area: (**a**) radar PPI screenshot, (**b**) radar signals including range-Doppler images and time-frequency data; (**c**) confirmation with EO/IR data; (**d**) flowchart of one collection.

The radar is an X-band pulse-Doppler phased array radar with a narrow band and a range resolution of approximately 12 m. The radar is equipped with an active electronically scanned phased array antenna and deployed on a rotating table to achieve 360° coverage in azimuth scans. The flexibly configured rotating speed is between 2 and 20 s. The radar is capable of tracking 1000 targets simultaneously. Equipped with the ATR function in real-time mode, it can recognize different targets from complex scenes, including birds, drones, vehicles, ships, people, and helicopters. The detection response time is nearly a millisecond. As such, it can present detection results with graphic icons, which label the recognition results, along with the rotating motion of the scanning beam, as shown in Figure 2a; the numbers around the icons are tracking numbers.

The most powerful application of this radar system is situational awareness using our patented classify while scan (CWS) technology. Figure 2a demonstrates the situational awareness of the surveillance area, where several ships, many birds, one helicopter, and our drone are active. The CWS module processes radar data in every radar cell and outputs the targets along with the ranged cells in the radar beam. The whole scenario is presented in a radar display following the scan of the radar beam. The radar beam continues to scan the area and capture the traces of the active targets. The dotted white lines represent the tracking paths of the objects. The tracking data provide the time situation analysis of each target, in addition to the 3D position.

## 3. Results

The tracking data indicate that TX25A moved faster than DJI Phantom 4. Figure 3 shows the tracking results of the two drones. In Figure 3a, the horizontal axis is the number of tracking frames and the vertical axis is the detection range, and the azimuth is shown in Figure 3b. The tracking range of TX25A and DJI Phantom 4 in this paper was from 10 to 12 km. The moving direction was away from the radar location; therefore, the Doppler speed is negative. The tracking period was approximately 10 s, and the sum of tracking frames for TX25A and DJI Phantom 4 is 12 and 22, respectively. Thus, TX25A flew approximately 1.8 km during 120 s, while DJI Phantom 4 flew the same distance in 220 s. Over this distance, the azimuth degrees of the two drones changed little. Therefore, the mean speed of TX25A, 16 m/s, is almost two times that of DJI Phantom 4, 8.5 m/s.

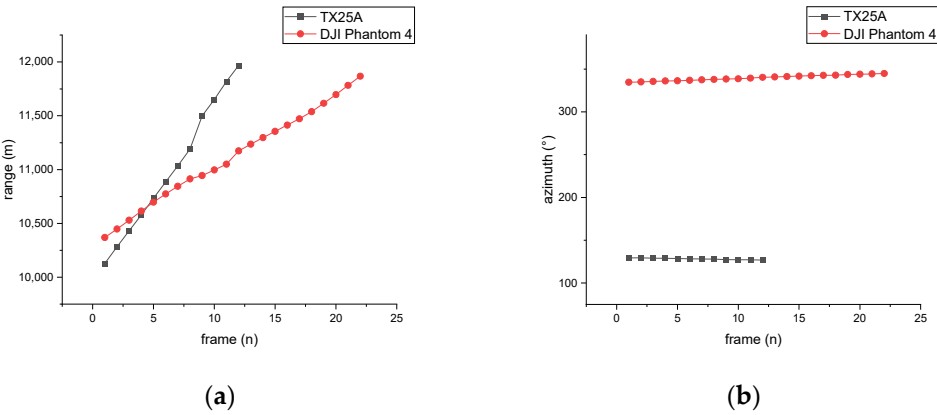

| (a) | (b) |

**Figure 3.** Detection of hybrid VTOL fixed-wing drone (TX25A) and quad-rotor drone (DJI Phantom 4) using a radar in a coastal area: (**a**) detection ranges; (**b**) azimuth degrees.

The measured RCS was larger for TX25A than DJI Phantom 4. Figure 4 shows the comparisons of signal amplitudes and signal-to-clutter (SCR) values of the two drones. Here, the signal amplitude means the peak number in the time series of radar echoes from the target in one radar cell. If there is calibration radar data of known RCS values of a standard object, the signal amplitude can be converted into RCS values of the measured target. Unfortunately, we do not have the calibration data; thus, we cannot present the RCS values here. Nevertheless, the signal amplitude is equivalent to the RCS value to some degree. The statistical data indicate that the mean signal amplitude of TX25A and DJI Phantom 4 was 235.62 and 98.96, respectively; the former is approximately 2.5 times the latter. In addition, in all sample frames of the same detection range, the mean signal amplitude was much larger for TX25A than DJI Phantom 4, meaning that the mean RCS value of TX25A was as expected, at least two times larger than that of DJI Phantom 4. In addition, the signal amplitude of the two drones fluctuated with the detection range, and the standard deviation of the signal amplitude was 119.75 for TX25A and 25 for DJI Phantom 4.

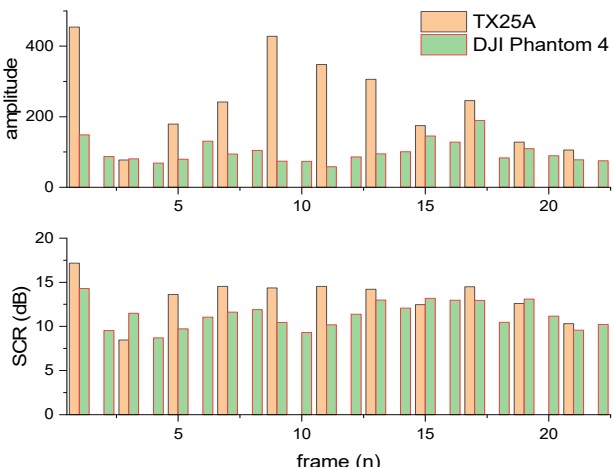

**Figure 4.** Comparisons of signal amplitudes and SCR values of VTOL and quad-rotor drones.

The measured SCR values were similar between TX25A and DJI Phantom 4. The SCR is defined as the ratio between the spectral peak and the spectrum's mean power; it is discussed in detail in our earlier work [41], and here we only refer to the definition. *SCR* is calculated by

$$SCR = \frac{F(D)}{\sum_1^N F(k)/N} \qquad (5)$$

where $F(k)$ is the amplitude of frequencies in the spectrum, $N$ is the length of the spectrum, and $D$ is the marked frequency of a target. Briefly, the *SCR* value indicates the radar cell's prioritized scattering power source. The mean *SCR* values of TX25A and DJI Phantom are 13.29 and 11.28 dB, respectively. The difference is only 2 dB, which is much smaller than the signal amplitudes. More importantly, the *SCR* values have a standard deviation of 2.15 and 1.44 for TX25A and DJI Phantom 4, respectively, indicating that they are much more stable than the signal amplitudes, which fluctuate highly with the detection range. Thus, *SCR* seems to be less affected by the detection range than signal amplitude or RCS.

JEM signatures appeared in radar echoes of both TX25A and DJI Phantom 4. Figure 5 demonstrates the usual JEM spectra of TX25A, DJI Phantom 4, and a fixed-wing drone, Albatross1, measured at ranges of 10 and 12 km. The time domain in one radar cell contains 96 points during a coherent pulse interval (CPI) of 20 ms, but its frequency domain is calculated with 256 points after zero padding. Most Doppler shifts of approximately 0 belong to sea clutter, and the green frames and circles in the figure highlight the JEM signatures in the four spectra. Albatross1 is a fixed-wing drone with one puller rotor. Its body size is 0.8 m and the blade length is 0.1 m. There are no JEM signals in the spectra of Albatross1. Each spectrum contains approximately three or four JEM-like peak envelopes, and they share a similar amplitude but have specific adjacent intervals. First, the JEM signature is independent of the detection range. As the detection range increases, the signal amplitude will decay, but the JEM signature stands still. Second, TX25A looks more like a fixed-wing drone than a multirotor drone, but its radar signals are similar to those of a multirotor drone because the four lifting blades produce modulation of the radar signals.

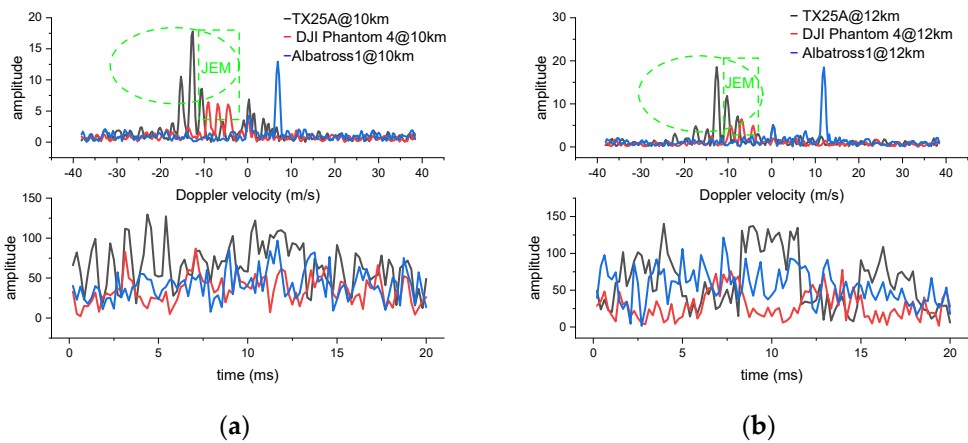

**Figure 5.** Raw radar data and spectra of different types of drones: (**a**) detection range of 10 km; (**b**) detection range of 12 km.

Details distinguish the different JEM spectra between TX25A and DJI Phantom 4, including magnitude, number, and spacing intervals. First, the absolute values of the JEM peaks are larger for TX25A than DJI Phantom 4 because the RCS value of TX25A is larger. Second, the Doppler speeds corresponding to the JEM peaks are higher for TX25A than DJI Phantom 4 because the body Doppler of TX25A is about two times faster, and micro-Doppler motions move based on body movement. Third, the JEM frequency spacing interval is wider for TX25A than DJI Phantom 4. The frequency resolution of the radar data shown in Figure 5 is approximately 20 Hz; thus, the JEM frequency spacing interval is approximately 160 Hz for TX25A and 320 Hz for DJI Phantom 4. Finally, according to Formula (3), the calculated rotary speed is approximately 40 revolutions per second (rps) for TX25A, and approximately 80 rps for DJI Phantom 4.

## 4. Discussion

JEM is independent of the detection range; thus, it can be a robust radar signature for ATR applications. Figure 6 demonstrates that there is always JEM in radar signals of TX25A from different detection ranges. The tracking ranges are from 8 to 13 km, and we selected data every 1 km (Figure 6). Some JEM signals have three peaks, and others have four peaks. Although the number and amplitude of JEM peaks in the spectrum are different, JEM appears in each spectrum. The red stars in the spectrum mark the highest Doppler; basically, they are body Dopplers. The statistical results of all radar echoes from TX25A indicate that the appearance probability of JEM is almost 100% in this coastal area using X-band radar. Therefore, if we detect JEM, we will know that the echoes come from an aircraft, not a bird.

Both TX25A and DJI Phantom 4 have fewer JEM peaks than the theoretical values. Since TX25A has five blades and DJI Phantom 4 has four blades, the theoretical Doppler numbers are six for TX25A (five JEM peaks and one body peak) and five for DJI Phantom (four JEM peaks and one body peak). However, the detected numbers shown in Figure 5 indicate that some JEM peaks were missing in the drones' spectra. The same situation is shown in the examples of TX25A in Figure 6, where we could not find one spectrum with the whole JEM.

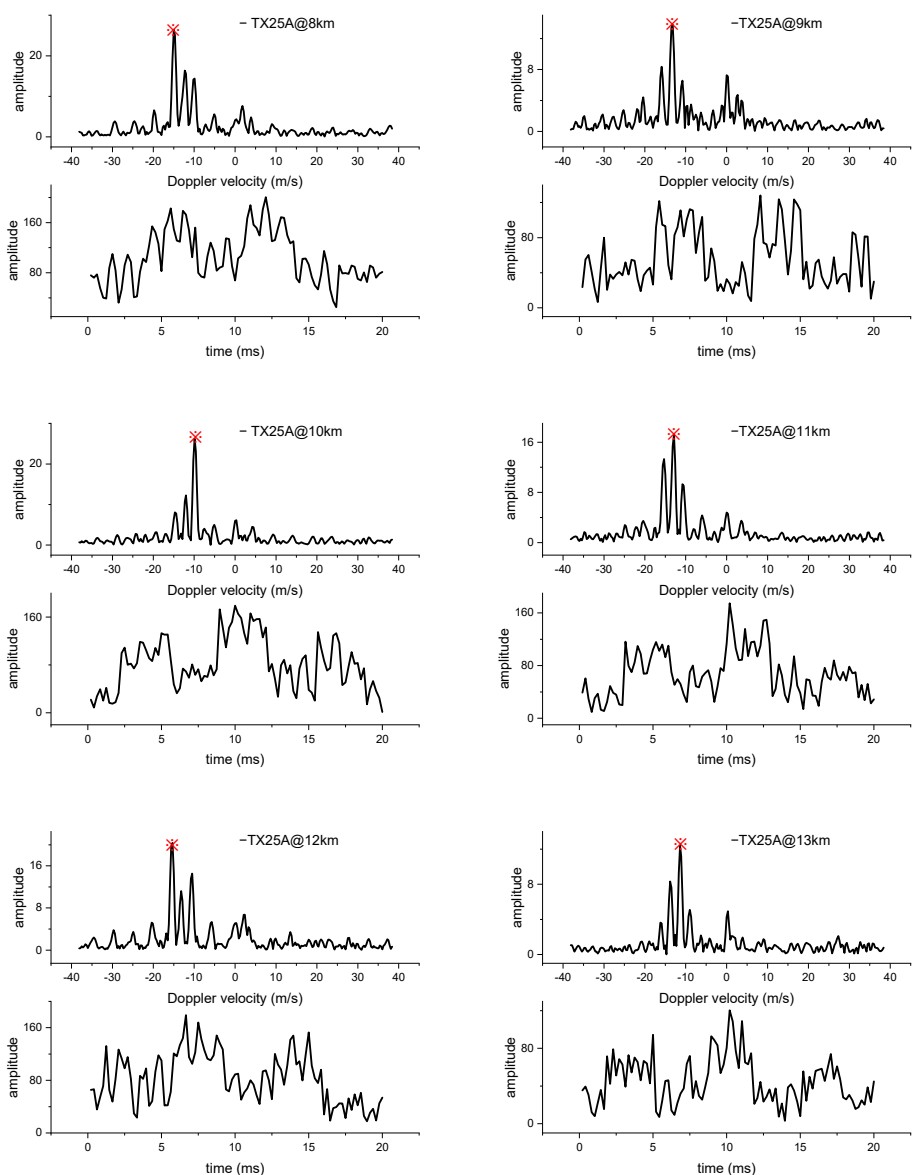

**Figure 6.** JEM of TX25A with different detection ranges.

Inadequate sampling time is the first reason for this finding. The radar dwell time (or measurement time) must include two rotation cycles of the blade to separate spool line spectra indicating one rotation frequency. For a rotation period of *T* s, the minimum dwell time is 2*T* s. However, the sampling time in Figure 5 is only 20 ms, much longer than the blade rotating period of either TX25A or DJI Phantom 4, which is at least 10 ms. Thus, sub-Nyquist sampling causes insufficient JEM signals containing fewer JEM peaks in the spectrum. In addition, the frequency resolution is limited to separating any micro-Doppler content. The rotating direction of lifting blades is equidirectional to body Doppler; therefore, the difference between micro-Doppler and body Doppler strengthens, resulting in the appearance from the span of body Doppler. However, since the rotating direction of TX25A's puller blades is perpendicular to the flight direction, the difference between the body Doppler and micro-Doppler of the puller blades is too small to stand out of the span of the body Doppler. Therefore, the JEM spectrum cannot always contain the puller blade Doppler.

The strongest Doppler in the spectrum does not come entirely from body movement. The Doppler peaks in the spectrum contain both body Doppler and micro-Doppler. Usually, the scattering power of the body is assumed to be stronger than the microstructure of the blades. As shown in Figure 5, the strongest peaks (about −12.6 m/s) in the spectra of TX25A come

from body movement, in reference to the flying velocity based on the GPS data. However, the situation is different with radar echoes from DJI Phantom. The three spectral peaks ($-9.3$, $-6.6$, $-4.2$ m/s) in Figure 5a are too similar to recognize the body Doppler ($-6.6$ m/s). Currently, the scattering power of the drone body ($-6.6$ m/s) is smaller than that of the rotating blades ($-9.3$ and $-4.2$ m/s); however, the scattering power of the body becomes larger than that of the rotating blades (Figure 5b). We note that the ratio of blade length to body size for TX25A and DJI Phantom 4 is 0.5 and 0.15, respectively. In addition, the blade material of TX25A (polycarbonate (PC)) is different from the body material (fiber-reinforced plastic (FRP)), while for DJI Phantom 4 they seem to be similar (PC). This exciting difference is relevant to the ratio between body size, blade size, and materials.

Since most tracking algorithms depend on Doppler estimation, JEM may mislead the extraction of body Doppler. In Figure 7, 1st Doppler means the strongest Doppler peak with the largest SCR (e.g., the Doppler marked with a red star in Figure 6), 2nd Doppler means the second strongest Doppler peak with the second largest SCR, and Mean Diff-Doppler represents the mean speed during the tracking period, which is related to the body-Doppler velocity. The 1st Doppler of TX25A seems to be closer to the Mean Diff-Speed than the 2nd Doppler. However, the body Doppler of DJI Phantom 4 may jump between 1st Doppler and 2nd Doppler. As such, the wrong tracking could be obtained. Without precise tracking, the effector unit of the counter-drone system could accidentally injure other objects, such as a bird or a person, resulting in terrible consequences; therefore, there is a need to classify JEM Doppler and body Doppler when dealing with radar tracking of targets such as drones.

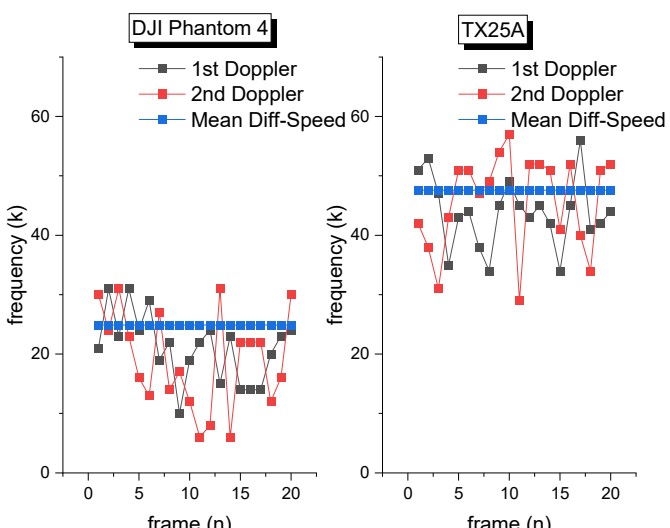

**Figure 7.** Tracking Doppler values of TX25A and DJI Phantom 4 drones.

## 5. Conclusions

While hybrid VTOL fixed-wing drones are new at present, they could become more popular in the future because of their ability to hover, like a multirotor drone, and their long endurance flight, like a fixed-wing drone. A VTOL drone looks more like a fixed-wing drone than a multirotor drone. As shown in Table 3, the investigation of radar signals of a typical VTOL fixed-wing drone, TX25A, and a quad-rotor drone, DJI Phantom 4, indicate that TX25A has twice the RCS value, twice the flying speed, and a 2 dB larger SCR value than DJI Phantom 4. Moreover, TX25A has JEM signals in its radar echoes, because the four lifting blades can modulate radar waves, similar to the radar signals of DJI Phantom 4. Yet, the Doppler produced by the puller blades seems to disappear in the spectrum because of insufficient sampling conditions and relative rotating direction. In addition, the measured JEM frequency spacing intervals of TX25A and DJI Phantom 4 could be robust ATR signatures. This research investigated the detection, classification, and tracking of new VTOL fixed-wing drones using radar systems and may inspire future research. One future research topic is to

design a signal processing algorithm to enhance the frequency resolution for visualizing the puller Doppler in the spectrum, making the puller Doppler a radar signature. If we can get the whole JEM, we could use it to classify the radar signals of both VTOL and quad-rotor drones, because the latter have no puller blades. Another topic to investigate is finding a method to classify the body Doppler and micro-Doppler in the spectra of drones, because we find that some micro-Doppler (or JEM) can suppress body Doppler, and developing an interference radar tracking algorithm based on extracting body Doppler.

**Table 3.** Comparison of radar signatures of drones.

| Contents | TX25A | DJI Phantom 4 |
|---|---|---|
| Body size | 0.40 m | 1.97 m |
| Signal amplitudes | 235.62 | 98.96 |
| SCR | 13.29 dB | 11.28 dB |
| Speed | 16 m/s | 8.5 m/s |
| Rotating rate | 40 rps | 80 rps |
| JEM peaks | 3 | 3 |

Note: Most of these are mean values.

**Author Contributions:** Conceptualization, J.G. and J.Y.; methodology, J.Y.; software, D.K.; validation, J.G. and J.Y.; formal analysis, J.G.; investigation, J.Y.; resources, D.L.; data curation, J.Y.; writing—original draft preparation, J.G.; writing—review and editing, H.H.; visualization, H.H.; supervision, J.Y.; project administration, D.L.; funding acquisition, D.K. All authors have read and agreed to the published version of the manuscript.

**Funding:** This research received no external funding.

**Institutional Review Board Statement:** Not applicable.

**Informed Consent Statement:** Not applicable.

**Data Availability Statement:** Some of the data presented in this study may be available on request from the corresponding author. The data are not publicly available due to the internal restriction of the research group.

**Acknowledgments:** We appreciate both the testers during the collection of the data, and we also want to thank to the authors whose photographs are reproduced in this study.

**Conflicts of Interest:** The authors declare that they have no conflict of interest.

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
