# Peer review of "Comparison of Radar Signatures from a Hybrid VTOL Fixed-Wing Drone and Quad-Rotor Drone"

_drones, doi:10.3390/drones6050110_

Round 1
Reviewer 1 Report
This manuscript ‘Comparison of radar signatures from a hybrid VTOL fixed-wing drone and a quad-rotor drone’ reviews the state-of-the-art in the X-band radar signatures of drones and provides a priori investigation of radar detecting, classifying, and tracking drones. Although this manuscript falls within the aim and scope of this journal, it needs to be revised mainly due to a lack of sufficient novelty. There are also some issues that should be addressed by the authors. Hence, a major revision of the paper is required. Detailed comments below.
GENERAL REMARKS
- Even the ‘Keywords’ has mistakes: VOTL. The paper should be revised thoroughly.
- The current abstract should be rephrased. A good abstract should accurately describe the background, motivation, methods, results, and conclusion of the manuscript in a clear and concise manner.
- In the introduction, the significance of the work carried out in this manuscript should be better explained relative to other recently published works. It is suggested to detail the main contributions of each reference and explain the differences between the present work and other published works.
- The literature review is severely lacking. To validate the improvement of this research, this manuscript should build the referenced previous method (based on the state-of-the-art research before this manuscript) for the comparison of the improvement of this research.
- My main concern with the manuscript is the absence of novelty of mathematical notations and equation presentations. Thus, the authors are requested to revise their manuscript to improve this methodological section.
- In the conclusion, it is suggested to summarize the highlights of investigation results that convey the key findings to the readers.
- The proof-reading acknowledgement should be specific to the scholars rather than to the company and it is recommended that the focus be on academic corrections. For example: acknowledgement - thanks Dr. XXXX/Prof. XXXX for the proofreading and correction.
SPECIFIC REMARKS
- Insufficient degree of innovation in equipment application and detection methods and lack of excavation of experimental results.
- The contributions of the conclusions limited; thus, more and deeper research is required.
- There are many mistakes: vertical take-off and landing (VTOL) (line 25, line 50, why two times), SMALL Drone (line 41), MICRO drones or MINI (line 46), etc.
STRONG POINTS:
- This research lifts the veil of detecting, classifying, and tracking some drones using radar systems.
WEAK POINTS:
- Limited novelty.
- Incomplete formulations.
- Incomplete literature review.
- Even the ‘Keywords’ includes mistakes.
Therefore, this manuscript should be re-reviewed if the authors can address the comments above.
Author Response
Open Review #1:
This manuscript ‘Comparison of radar signatures from a hybrid VTOL fixed-wing drone and a quad-rotor drone’ reviews the state-of-the-art in the X-band radar signatures of drones and provides a priori investigation of radar detecting, classifying, and tracking drones. Although this manuscript falls within the aim and scope of this journal, it needs to be revised mainly due to a lack of sufficient novelty. There are also some issues that should be addressed by the authors. Hence, a major revision of the paper is required. Detailed comments below.
Response: Thanks for your comments. We will revise the paper, after we consider reviewers’ advices.
GENERAL REMARKS
- Even the ‘Keywords’ has mistakes: VOTL. The paper should be revised thoroughly.
Response: Thanks for your advice, and we have revised the paper with the help of the editing services provided by the MDPI.
- The current abstract should be rephrased. A good abstract should accurately describe the background, motivation, methods, results, and conclusion of the manuscript in a clear and concise manner.
Response: The abstract has been revised after we refer to your advice.
- In the introduction, the significance of the work carried out in this manuscript should be better explained relative to other recently published works. It is suggested to detail the main contributions of each reference and explain the differences between the present work and other published works.
- The literature review is severely lacking. To validate the improvement of this research, this manuscript should build the referenced previous method (based on the state-of-the-art research before this manuscript) for the comparison of the improvement of this research.
Response to 3&4 : Thanks for your advice. We revise the introduction, but as we propose in the paper that since little references have mentioned to radar detection of VTOL drones, and then the only differences between the present works and our work in this paper could be that they investigated radar signals of quad-rotor/ multi-rotor drones, but we detected radar echoes from a new type drone, i.e., VTOL drones. Thereby, we re-state this difference in the introduction.
- My main concern with the manuscript is the absence of novelty of mathematical notations and equation presentations. Thus, the authors are requested to revise their manuscript to improve this methodological section.
Response : Thanks for your comment. W still revise the Section Method & Materials. Firstly, we move the content dealing with radar and test environment from Section Result to Section Method. Secondly, we supply the flow chart when detecting the radar signals. Thirdly, we organize the contents in Section Method, and divide them in three parts, including (1) introducing the tested drone, (2) analyzing the radar signatures, (3) discussing the radar and test environment. Thus, the methodological section in Section Method is more reasonable.
- In the conclusion, it is suggested to summarize the highlights of investigation results that convey the key findings to the readers.
Response: Thanks for your advice, and we highlight the results by state the results in Table 3 in Section Conclusion.
- The proof-reading acknowledgement should be specific to the scholars rather than to the company and it is recommended that the focus be on academic corrections. For example: acknowledgement - thanks Dr. XXXX/Prof. XXXX for the proofreading and correction.
Response: Thanks for your advice, and we revise this part following your suggestion.
SPECIFIC REMARKS
- Insufficient degree of innovation in equipment application and detection methods and lack of excavation of experimental results.
- The contributions of the conclusions limited; thus, more and deeper research is required.
- There are many mistakes: vertical take-off and landing (VTOL) (line 25, line 50, why two times), SMALL Drone (line 41), MICRO drones or MINI (line 46), etc.
STRONG POINTS:
- This research lifts the veil of detecting, classifying, and tracking some drones using radar systems.
WEAK POINTS:
- Limited novelty.
- Incomplete formulations.
- Incomplete literature review.
- Even the ‘Keywords’ includes mistakes.
Therefore, this manuscript should be re-reviewed if the authors can address the comments above.
Response to SPECIFIC REMARKS & STRONG POINTS & WEAK POINTS: Thanks for your comments. The most significance of the contents in this paper is the investigation of radar signatures of the VTOL drones. Most radar signatures include RCS, Doppler velocity, micro-Doppler, trace, and more. In addition to the above signatures, we also analyze the SCR values of the VTOL drone & quad-rotor drone, and the specifical micro-Doppler (i.e., JEM) produced by the rotating blades. The measured results are shown in corresponding figures with explanations, including Doppler velocity in Fig. 3, RCS & SCR in Fig. 4, and JEM in Fig. 5. The measured radar signatures of VTOL could be used for ATR applications. Yet, as we found in Fig. 5&6, not all JEM Dopplers produced by all rotating blades including four lifting blades and one puller blade can appear in the spectrum. There is a difference between the theoretical ones and the test ones, that is caused by insufficient JEM signals. Besides, JEM Doppler may be a useful signature but also an interference for radar tracking algorithm based on extracting body Doppler because they suppress the body Doppler in the spectrum. Thereby, as we state in Section Conclusion, there are at least two research topics to do in the future. One is to design algorithm to enhance the JEM and appear the missing Doppler produced by the puller blade. If we get the whole JEM, we may use it to classify the radar signals of VTOL drones and the quad-rotor drones, because quad-rotor drones have no puller blades. Another topic is to find a method to separate the body Doppler and the micro-Doppler in the spectrum for radar tracking algorithm, otherwise, the radar track using wrong Doppler could be wrong. We are currently busy with the two topics, and hope find solutions in the future.
This paper is not a usual paper dealing with some an algorithm, but we discuss the scattering phenomenon (or radar signatures) of a new trendy VTOL drone. Thereby, we believe the innovation of this paper are the first presentation of those radar signatures of VTOL drones.

Reviewer 2 Report
The authors presented an investigation of radar detecting, classifying, and tracking new VTOL fixed-wing drones. Specifically, the authors introduced the characteristics of a VTOL fixed-wing drone, TX25A, and a multirotor drone, DJI Phantom 4, and conducted the radar signature comparison. Overall, the results support the discussion and conclusion. However, there are a few issues with this study.
- The main motivation of this study is that current studies mention little about radar signatures of hybrid VTOL fixed-wing drones. To some extent, it's interesting to the readers. But the authors simply briefly described the characteristics of the drones and conducted the radar signature comparison. The current studies are too limited to show the significance, it lacks significance and novelty.
- The authors mentioned that the current research lifts the veil of detecting, classifying, and tracking new VTOL fixed-wing drones using radar systems and may inspire future research. I agree. However, more discussion and testing should be given to support this claim. Given that the current content is limited, adding more tests to show the future work possibility might increase the overall significance.
- The acronym should be given the full name the first times they are used.
- In the Introduction section, it's not clear what is the significance of this study.
- Line 50, the author mentioned that "This paper probably demonstrates the first radar echoes of a hybrid vertical takeoff and landing (VTOL) fixed-wing drone...". Please make sure whether this work is the first study on VTOL or not, and don't use 'Probably'.
- A brief description of TX25A and DJI Phantom 4 should be given in the Introduction section before they are used.
- There are some typos and grammar issues, please fix them. Eg. Figure 1 title.
Author Response
Open Review #2:
The authors presented an investigation of radar detecting, classifying, and tracking new VTOL fixed-wing drones. Specifically, the authors introduced the characteristics of a VTOL fixed-wing drone, TX25A, and a multirotor drone, DJI Phantom 4, and conducted the radar signature comparison. Overall, the results support the discussion and conclusion. However, there are a few issues with this study.
Response: Thanks for your comments. We will revise the issues and present a better paper.
- The main motivation of this study is that current studies mention little about radar signatures of hybrid VTOL fixed-wing drones. To some extent, it's interesting to the readers. But the authors simply briefly described the characteristics of the drones and conducted the radar signature comparison. The current studies are too limited to show the significance, it lacks significance and novelty.
Response: Thanks for your advice. We believe the significance is the presentation of these radar signatures of the VTOL drones. Before we collect radar signals of the VTOL drone, we thought that the radar signals could be more similar to that of a fixed-wing drone. But the measurement results (Fig. 5) indicate that radar signals of a VTOL drone look more like a quad-rotor drone than a fixed-wing drone. Furthermore, we also note the missing Doppler produced by the lifting blade of the VTOL drone, and the interference to the body Doppler caused by the JEM Doppler. Thereby, on one hand, these radar signatures presented in the paper could be useful in ATR applications. On the other hand, these findings about the missing Doppler and the interference could inspire other research topics. In fact, we are currently investigating methods to solve the two problems.
- The authors mentioned that the current research lifts the veil of detecting, classifying, and tracking new VTOL fixed-wing drones using radar systems and may inspire future research. I agree. However, more discussion and testing should be given to support this claim. Given that the current content is limited, adding more tests to show the future work possibility might increase the overall significance.
Response: Thanks for your advice, and we add some test results to support our claims.
- The acronym should be given the full name the first times they are used.
Response: Thanks for your advice, and we will revise it, as well as other language problems.
- In the Introduction section, it's not clear what is the significance of this study.
Response: Thanks for your comment. We revise the introduction section. Basically, the section presents the fact that current researches mention little radar detection of VTOL drones, and then introduce that we will investigate the radar detection of a VTOL drone. In fact, since the VTOL drone is a new trend, there is obvious no much radar detection of such VTOL drones. Note that, the VTOL drone means the VTOL drone looks similar to TX25A which is an industry-type drone, not the tiny drone like that in [21].
- Line 50, the author mentioned that "This paper probably demonstrates the first radar echoes of a hybrid vertical takeoff and landing (VTOL) fixed-wing drone...". Please make sure whether this work is the first study on VTOL or not, and don't use 'Probably'.
Response: Thanks for your comment. As far as we know, this paper demonstrates the first radar detection of such VTOL drone which looks like the TX25A. Yet, since the VTOL drones contains many different types of VTOL types. For examples, the VTOL drones with four lifting motors (e.g., our TX25A), and that with two lifting motors (e.g., that in [21]). Thereby, we revise the sentence with more strict adjective words.
- A brief description of TX25A and DJI Phantom 4 should be given in the Introduction section before they are used.
Response: Thanks for your comment, and we will add some brief description in introduction section.
- There are some typos and grammar issues, please fix them. Eg. Figure 1 title.
Response: Thanks for your comment, and we have revised the paper with the help of the editing services provided by the MDPI.

Reviewer 3 Report
The article is very interesting and addresses a current topic evaluating radar effectiveness on UAV objects of various designs. The research problem presented in the article is little reported in the literature and is an interesting research task. Amalia and layout of the article is correct. However, I request that a few corrections be made:
Please add more references to existing literature because citations are too few.
Please give technical parameters of tested UAVs
Please describe presented results in detail and add more in conclusions about further work
Author Response
Open Review #3:
The article is very interesting and addresses a current topic evaluating radar effectiveness on UAV objects of various designs. The research problem presented in the article is little reported in the literature and is an interesting research task. Amalia and layout of the article is correct. However, I request that a few corrections be made:
Response: Thanks for your comment, and we will revise the paper by referring to your advice and others.
Please add more references to existing literature because citations are too few.
Response: Thanks for your comment, we add more references. As we stated in the paper, since we investigate the radar signatures of such a new VTOL drone, there are little available researches once discuss the topic. Yet, we still add some references dealing with radar detection of quad-rotor drones.
Please give technical parameters of tested UAVs
Response: Thanks for your comment, but we believe the technical parameters of tested UAVs are presented in Table 2.
Please describe presented results in detail and add more in conclusions about further work
Response: Thanks for your comment. We revise the measurement results in Table 3, and present more details in conclusion. Moreover, we also provide the two topics in future work. One is to find a method to enhance the JEM and make the Doppler produced by the puller rotor appear in the spectrum. The other one is to investigate a solution to separate body Doppler from the micro-Doppler in the spectrum for improving the radar tracking algorithm. In fact, we are currently doing the researches to find solutions.

Round 2
Reviewer 1 Report
Greatly improved. There are minor grammar weaknesses to resolve with a final proofread
Reviewer 2 Report
The author has made corresponding changes.